# First-Principles Study of Thermo-Physical Properties of Pu-Containing Gd_2_Zr_2_O_7_

**DOI:** 10.3390/nano9020196

**Published:** 2019-02-03

**Authors:** Pengcheng Li, Fengai Zhao, Haiyan Xiao, Haibin Zhang, Hengfeng Gong, Sa Zhang, Zijiang Liu, Xiaotao Zu

**Affiliations:** 1School of Physics, University of Electronic Science and Technology of China, Chengdu 610054, China; pengchengli257335@gmail.com (P.L.); 15136231837@163.com (S.Z.); xtzu@uestc.edu.cn (X.Z.); 2Institute of Fundamental and Frontier Sciences, University of Electronic Science and Technology of China, Chengdu 610054, China; fengaizh0506@163.com; 3Institute of Nuclear Physics and Chemistry, Chinese Academy of Engineering Physics, Mianyang 621900, China; 4Department of ATF R&D, China Nuclear Power Technology Research Institute Co., Ltd., Shenzhen 518000, China; gonghengfeng@cgnpc.com.cn; 5Department of Physics, Lanzhou City University, Lanzhou 730070, China; lzjcaep@126.com

**Keywords:** DFT+U, Gd_2_Zr_2_O_7_, nuclear waste, mechanical properties

## Abstract

A density functional theory plus Hubbard U method is used to investigate how the incorporation of Pu waste into Gd_2_Zr_2_O_7_ pyrochlore influences its thermo-physical properties. It is found that immobilization of Pu at Gd-site of Gd_2_Zr_2_O_7_ has minor effects on the mechanical and thermal properties, whereas substitution of Pu for Zr-site results in remarkable influences on the structural parameters, elastic moduli, elastic isotropy, Debye temperature and electronic structure. The discrepancy in thermo-physical properties between Gd_2−y_Pu_y_Zr_2_O_7_ and Gd_2_Zr_2−y_Pu_y_O_7_ may be a result of their different structural and electronic structures. This study provides a direct insight into the thermo-physical properties of Pu-containing Gd_2_Zr_2_O_7_, which will be important for further investigation of nuclear waste immobilization by pyrochlores.

## 1. Introduction

As the nuclear industry develops fast, ways to treat spent fuel and nuclear waste safely, such as plutonium and minor actinides (Np, Am, Cm), has become an important environmental conservation issue [1,2,3]. It is acceptable to store spent fuel and separated waste in stainless steel vessels in the short term, but in the long term it is hoped that this material will be transformed into more secure and manageable solids [1,4,5]. One method proposed for the treatment of plutonium is immobilization in zirconate pyrochlores, particularly Gd_2_Zr_2_O_7,_ which has high thermal stability, high chemical durability, and high radiation tolerance [6,7,8,9]. Besides, Gd is an effective neutron absorber [6].

Experimentally, Pu is often substituted by nonradioactive cerium (Ce), since they share the same crystallographic structure, and thermo-physical and chemical properties [10,11]. Zhao et al. synthesized (Gd_1−x_Ce_x_)_2_Zr_2_O_7+x_ (0 ≤ x ≤0.6) solid solutions, indicating that Ce^3+^ ions can be incorporated into the Gd^3+^ sites. They proposed that the content of ${\mathrm{Pu}}^{3+}$ immobilized at Gd-site was less than 40 mol% [12]. On the other hand, Gd_2_(Zr_1−x_Ce_x_)_2_O_7_ (0≤ x ≤1.0) solid solutions have been synthesized by Reid et al. [13] and Patwe et al. [14]. They found that Ce can be immobilized at Zr sites entirely as Ce^4+^, which leads to structural transformation from pyrochlore to fluorite phase, and its composition ranges from Gd_2_Ce_0.2_Zr_1.8_O_7_ to Gd_2_Ce_1.7_Zr_0.3_O_7_. Similar solution behavior and structural properties for Pu incorporation in Gd_2_Zr_2_O_7_ have been obtained by first-principles calculations [15,16,17]. However, different electronic structures can be obtained for Ce and Pu immobilization at the Gd-site of Gd_2_Zr_2_O_7_, i.e., the band gap increases and reduces when Ce and Pu substitutes for Gd site, respectively [15,18]. These differences mainly result from the different <Ce-O> and <Pu-O> interactions at band edges. The different electronic structure may result in varying mechanical properties. For Young’s modulus, which is described by $E\;\propto \;\frac{M_{a}}{r_{0}{}^{4}},$ with $M_{a}$ the Madelung constant and r_0_ the interionic spacing [19], it is very sensitive to r_0_. For ionic crystals, the r_0_ is affected by bond interactions. According to these analyses, large discrepancies in electronic structures between Ce and Pu incorporation in Gd_2_Zr_2_O_7_ may lead to different Young’s modulus. This indicates that for the mechanical properties of pyrochlores, Ce may not be a good substitute for Pu, despite the two having several similar thermo-physical properties. Thus far, there are no reports on the mechanical properties of Pu immobilization in Gd_2_Zr_2_O_7_. It is necessary to explore how Pu doping influences the mechanical properties of Gd_2_Zr_2_O_7_, because the knowledge of thermo-mechanical characteristics, for example, elastic moduli and Debye temperature is important for safe fuel disposal [20]. It provides new perspectives into the behavior of actinide incorporation in pyrochlores for their applications in harsh environments.

In this work, the structural, mechanical and electronic properties of Pu incorporation in Gd_2_Zr_2_O_7_ are investigated by the density functional theory plus the Hubbard U method (DFT+U). The remaining part of the paper is structured as follows: Section 2 lists computational details; Section 3 contains our results and discussions, involving the structural stability, elastic constants with elastic moduli of Gd_2−y_Pu_y_Zr_2_O_7_ and Gd_2_Zr_2−y_Pu_y_O_7_, as well as the ductility, elastic anisotropy, Debye temperature and electronic structures of Pu-doped Gd_2_Zr_2_O_7_. In Section 4, we summarize our conclusions.

## 2. Computational Details

The density functional theory method within the Vienna *Ab-initio* Simulation Package (VASP) [21,22] are employed in all the computations. The Hubbard U correction [23] is considered to take into account the strong correlation interaction between 5f electrons of Pu and the effective U values are taken to be 4 eV. The projector augment wave (PAW) method [24] is used to describe the interaction between electrons and ions. As for the exchange-correlation functional, a number of generalized gradient approximation (GGA) functionals have been reported in the literature [25,26,27] and the functional parametrized by Perdew, Burke and Ernzerhof [28] is employed in this work. In the calculations, a 2 × 2 × 2 k-point sampling in reciprocal space is employed, with a cutoff energy of 600 eV for the plane wave basis sets. Figure 1 illustrates the schematic view of geometrical structure for the considered compounds, i.e., Gd_2−y_Pu_y_Zr_2_O_7_ and Gd_2_Zr_2−y_Pu_y_O_7_ (y = 0, 0.5, 1.0, 1.5, 2.0). The special quasi-random structure method is used to build the structural models for Pu immobilization at Gd-site and Zr-site [29,30,31,32]. 

## 3. Results and Discussion

### 3.1. Structural Stability of Pu Incorporation into Gd_2_Zr_2_O_7_

As Pu substitutes for Gd^3+^ and Zr^4+^ in Gd_2_Zr_2_O_7_, the corresponding valence states for Pu are Pu^3+^ and Pu^4+^, respectively. Because in both PuO_2_ and Pu_2_O_3_ the Pu 5f electrons are strongly correlated, Hubbard U correction is thus necessary. In the revised manuscript, we present the density of state distribution for both PuO_2_ and Pu_2_O_3_ at U_eff_ = 0 eV and U_eff_ = 4 eV in Figure 2. It is shown that without Hubbard U correction, i.e., at U_eff_ = 0 eV, the Pu 5f electrons are itinerant and delocalized over the Fermi lever, resulting in metallic states. At U_eff_ = 4 eV, the Pu 5f electrons are localized and the system becomes insulating, which is consistent with the experimental finding [32]. The calculated lattice constant of 5.46 Å for PuO_2_ and 11.18 Å for Pu_2_O_3_ obtained at U_eff_ = 4 eV are comparable to the experimental values of 5.39 Å [33] and 10.98 Å [34], respectively. The calculated band gap for Pu_2_O_3_ at U_eff_ = 4 eV is 1.757 eV, which corresponds to the experimental value of 2 eV [35]. Thus, we use U_eff_ = 4 eV in our subsequent calculations for Pu immobilization in Gd_2_Zr_2_O_7_. On the other hand, the 4 eV for U_eff_ is also consistent with the value of 4–5 eV that are reported in the literature [36,37].

A structural optimization is first performed for both Gd_2−y_Pu_y_Zr_2_O_7_ and Gd_2_Zr_2−y_Pu_y_O_7_. The calculated lattice constants, oxygen positional parameter $x_{O48f}$ and bond distances for Gd_2−y_Pu_y_Zr_2_O_7_ and Gd_2_Zr_2−y_Pu_y_O_7_ are listed in Table 1 and Table 2, respectively. The changes of lattice constant and $x_{O48f}$ for Gd_2−y_Pu_y_Zr_2_O_7_ and Gd_2_Zr_2−y_Pu_y_O_7_ with Pu concentrations are shown in Figure 3. The calculated lattice constant of 10.666 Å for Gd_2_Zr_2_O_7_ is slightly larger than the experimental value of 10.54 Å [38], while consistent with other calculations of 10.66 Å [18]. The calculated a_0_ of 10.802 Å for Pu_2_Zr_2_O_7_ is comparable with the experimental value of 10.70 Å [39]. As the Pu content increases, the lattice constant gradually increases for both Gd_2−y_Pu_y_Zr_2_O_7_ and Gd_2_Zr_2−y_Pu_y_O_7_, and it changes more significantly for Gd_2_Zr_2−y_Pu_y_O_7_ than that for Gd_2−y_Pu_y_Zr_2_O_7_. This is caused by the fact that the effective ionic radius of 1.053 Å [40] for Gd^3^ is in good agreement with the value of ^+^~1.1 Å [40] for Pu^3+^, but the effective ionic radius of 0.72 Å [40] for Zr^4+^ is much smaller than the value of 0.96 Å [40] for Pu^4+^. With regard to oxygen positional parameter $x_{O48f}$, the calculated value of 0.339 for Gd_2_Zr_2_O_7_ is smaller than the experimental value of 0.345 [41], and is comparable to the calculated value of 0.339 reported by Wang et al. [18]. For Gd_2−y_Pu_y_Zr_2_O_7_, the $x_{O48f}$ changes slightly as the Pu content increases, which indicates that the Gd_2−y_Pu_y_Zr_2_O_7_ remains the pyrochlore structure. Wang et al. [18] has observed similar phenomenon for Gd_2−y_Ce_y_Zr_2_O_7_. For Gd_2_Zr_2−y_Pu_y_O_7_, we find that the $x_{O48f}$ increases a lot, varying from 0.339 to 0.350, suggesting that the Gd_2_Zr_2−y_Pu_y_O_7_ tends to be a defect fluorite structure as the Pu content increases [15]. Comparing the lattice constant and oxygen positional parameter for Gd_2−y_Pu_y_Zr_2_O_7_ and Gd_2_Zr_2−y_Pu_y_O_7_, we find that the formation of Gd_2_Zr_2_O_7_-Pu_2_Zr_2_O_7_ solid solution is more preferable than that of Gd_2_Zr_2_O_7_ and Gd_2_Pu_2_O_7_. As for bond distances, the calculated <Gd-O_48f_> distance of 2.553 Å in Gd_2_Zr_2_O_7_ is a little larger than the experimental value of 2.483 Å [38], and is comparable to other calculated value of 2.548 Å [18]. Meanwhile, the calculated value of 2.109 Å for <Zr-O_48f_> distance is consistent with the experimental value [38] and other calculated value [18] of 2.110 Å. For Gd_2−y_Pu_y_Zr_2_O_7_, the <Gd-O_48f_> and <Pu-O_48f_> distances increase slightly and the <Gd-O_8b_> and <Pu-O_8b_> distances decrease slightly as the Pu content increases. Comparing the <Gd-O> and <Pu-O> bonds, we find that the <Pu-O_48f_> and <Pu-O_8b_> distances are slightly larger than <Gd-O_48f_> and <Gd-O_8b_> distances, respectively, i.e., Pu substitution for Gd-site leads to small increase in the bonding distance. For Gd_2_Zr_2−y_Pu_y_O_7_, the situation is different. The <Pu-O_48f_> bond is about 0.12–0.19 Å larger than the <Zr-O_48f_> bond. Simultaneously, the <Gd-O_8b_> bond increases a little as the Pu content increases. Consequently, there is a remarkable increase in the lattice constant of Gd_2−y_Pu_y_Zr_2_O_7_.

### 3.2. Elastic Constants and Elastic Moduli of Gd_2−y_Pu_y_Zr_2_O_7_ and Gd_2_Zr_2−y_Pu_y_O_7_

Elastic constants are response functions to the external forces and are very important in the materials’ properties [42]. Table 3 lists the calculated elastic constants along with available experimental and theoretical values. For Gd_2_Zr_2_O_7_, the calculated C_11_, C_12_ and C_44_ are 285.1, 102.5 and 82.1 GPa, respectively, showing good agreement with other calculations [43]. For Pu_2_Zr_2_O_7_, the calculated C_11_ = 270.6 GPa, C_12_ = 107.3 GPa and C_44_ = 81.2 GPa differ from reference [2], in which different calculational parameters are employed. It is noted that the elastic stability criteria are satisfied for all the investigated systems, i.e., C_11_ > |C_12_|, C_44_ > 0, and (C_11_ + 2C_12_) > 0 [44], implying that they are all mechanically stable. 

Figure 4 presents the changes of elastic constants with Pu content for both Gd_2−y_Pu_y_Zr_2_O_7_ and Gd_2_Zr_2−y_Pu_y_O_7_. For Gd_2−y_Pu_y_Zr_2_O_7_, as the Pu content increases, the elastic constants are affected minorly. As the Pu concentration increases, the C_11_ and C_12_ decreases and increases slightly, respectively, and the change in C_44_, is nearly negligible. As for Gd_2_Zr_2−y_Pu_y_O_7_, the variation of elastic constants with Pu content is more considerable. As the y value changes, the C_11_ and C_12_ first decreases, then increases, and finally decreases again. As for C_44_, it first decreases to y = 1.5, and then increases. Generally speaking, as the Pu content increases, there are more significant changes on Zr-site than Gd-site, meaning that Pu immobilization at Zr-site leads to remarkable variations in the mechanical properties of Gd_2_Zr_2_O_7_. Zhao et al. [47] also reported that Nd substitution of Zr-site of Gd_2_Zr_2_O_7_ greatly affects the mechanical properties.

From the calculated elastic constants, the elastic moduli, including the bulk modulus (*B*), shear modulus (*G*) and Young’s modulus (*E*), can be deduced [48,49,50,51], i.e.,
$$B=B^{V}=B^{R}=\frac{C_{11}+2C_{12}}{3},\;G^{V}=\frac{C_{11}-C_{12}+3C_{44}}{5},\;G^{R}=\frac{5\left(C_{11}-C_{12}\right)C_{44}}{4C_{44}+3\left(C_{11}-C_{12}\right)},\;G=G^{VRH}=\frac{G^{V}+G^{R}}{2},\;E=\frac{9BG}{3B+G}.$$

Here, the Voigt and Reuss evaluations for *B* and *G* are represented by *V* and *R*, respectively. Table 3 lists the calculated *B*, *G*, *E*, and others’ theoretical and experimental values for both Gd_2−y_Pu_y_Zr_2_O_7_ and Gd_2_Zr_2−y_Pu_y_O_7_. For Gd_2_Zr_2_O_7_, the calculated *B* = 163.4 GPa, *G* = 85.7 GPa, *E* = 218.8 GPa are comparable with experimental [20,45,46] and other calculated [43] results. Figure 5 shows the variation of elastic moduli for both Gd_2−y_Pu_y_Zr_2_O_7_ and Gd_2_Zr_2−y_Pu_y_O_7_. For Gd_2−y_Pu_y_Zr_2_O_7_, the elastic moduli change very slightly as the Pu content increases. When Pu is immobilized at Zr-site, there are remarkable changes. The bulk modulus first decreases, reaching a minimum of 134.3 GPa at y = 1.0, then rises up to 142.3 GPa at y = 1.5 and finally decreases to 136.9 GPa at y = 2.0. The shear modulus decreases to 58.9 GPa at y = 1.0 and increases slightly to 63.6 GPa at y = 2.0. The Young’s modulus decreases sharply from 218.8 GPa to 154.2 GPa as the y varies from 0 to 1.0, but increases again to 165.7 GPa at y = 1.5, finally changing slightly. The <Zr-O> bonds determine the total stiffness of A_2_Zr_2_O_7_ pyrochlore, because the corner-sharing ZrO_6_ octahedra constitutes its backbone, and the A^3+^ fills the interstices [19,52]. Therefore, the substitution of Zr^4+^ by Pu^4+^ causes the change of <Zr-O> bonds to <Pu-O> bonds and influences the Young’s modulus, especially for these ionic bonds [19,52]. The Young’s modulus E is described by $E\propto \frac{M_{a}}{r_{0}{}^{4}}$ for ionic bonds, in which M_a_ represents the Madelung constant and r_o_ represents the interionic distance [19]. The <Zr-O> bonds in Gd_2_Zr_2_O_7_ are affected little by Pu substitution for Gd-site, leading to slight effects on the Young’s modulus. For Gd_2_Zr_2−y_Pu_y_O_7_, the <Zr-O_48f_> bond length of 2.11 Å is smaller than the value of 2.26 Å for <Pu-O_48f_>, resulting in remarkable effects on the Young’s modulus. The bulk modulus, shear modulus and Young’s modulus for Nd doping of Gd_2_Zr_2_O_7_ have been calculated by Zhao et al. [47], who also reported that Nd immobilization at Zr-site has more remarkable influences on the elastic moduli than that at Gd-site.

For each ion in Gd_2−y_Pu_y_Zr_2_O_7_ and Gd_2_Zr_2−y_Pu_y_O_7_, we analyze the Bader charge to explore the origin of the discrepancy in the elastic moduli between Gd_2−y_Pu_y_Zr_2_O_7_ and Gd_2_Zr_2−y_Pu_y_O_7_. The Bader charge values are listed in Table 4. As Pu is immobilized at Gd-site and Zr-site, the average Bader charge for Pu are 2.10 |e| and 2.33 |e|, respectively, corresponding to the nominal +3 |e| and +4 |e| in oversimplified classical model [2]. For Gd_2−y_Pu_y_Zr_2_O_7_, the Bader charge for Gd and Pu ions are similar to each other, i.e., 2.15 and 2.10|e|, respectively. Wang et al. [18] calculated the Bader charge of Gd_2−y_Ce_y_Zr_2_O_7_ and reported very similar results. Additionally, the bonding distance for <Gd-O_48f_> and <Gd-O_8b_> are determined to be 2.57 Å and 2.30 Å, which are comparable to the values of 2.60 Å for <Pu-O_48f_> and 2.36 Å for <Pu-O_8b_>, respectively. Obviously, the <Gd-O> and <Pu-O> bonding interaction are very similar to each other, which explains why the mechanical properties of Gd_2_Zr_2_O_7_ are affected slightly by Pu immobilization at Gd-site. For Gd_2_Zr_2−y_Pu_y_O_7_, the situation is much different. The average Bader charge are 2.33 |e| for Pu ions and 2.26 |e| for Zr ions. Considering that the <Pu-O_48f_> distance of 2.26 Å is larger than the <Zr-O_48f_> distance of 2.11 Å and the ionic radius of 0.96 Å for Pu ions is larger than that of 0.72 Å for Zr ions [40], it is suggested that the <Zr-O> bonds exhibit weaker ionicity than <Pu-O> bonds in Gd_2_Zr_2−y_Pu_y_O_7_. Because of the brittleness of the ionic bonds, the immobilization of Pu at Zr sites will thus increase the ionicity and decrease the elastic moduli. 

### 3.3. Ductility, Elastic Anisotropy, Debye Temperature and Electronic Structures of Pu-Doped Gd_2_Zr_2_O_7_

Pugh’s indicator ($\frac{B}{G}$) is used to reflect the ductility of materials. If $\frac{B}{G}>1.75$, the material shows ductility; or, it is brittle [54]. Table 5 presents the calculated Pugh’s indicators. For Gd_2_Zr_2_O_7_, our value of 1.907 is comparable with the experimental values of 1.913 [45,46], 1.891 [20] and other calculated value of 2.004 [53]. For both Gd_2−y_Pu_y_Zr_2_O_7_ and Gd_2_Zr_2−y_Pu_y_O_7_, our calculations show that the Pugh’s indicators are all larger than 1.75, implying that all the considered composites are ductile. Poisson’s indicator (ν) can be employed to evaluate the relative ductility of materials. When ν is around 0.1, the material shows brittle covalent properties. When ν is bigger than 0.25, it exhibits ductile ionic properties [55]. Table 5 lists the calculated, experimental and others’ calculated Poisson’s ratio for both Gd_2−y_Pu_y_Zr_2_O_7_ and Gd_2_Zr_2−y_Pu_y_O_7_. For Gd_2_Zr_2_O_7_, the calculated Poisson’s ratio of 0.277 could be comparable to the experimental values of 0.276 [45,46], 0.274 [20] and other calculated results of 0.286 [53] and 0.273 [43]. The Poisson’s ratios for Gd_2−y_Pu_y_Zr_2_O_7_ and Gd_2_Zr_2−y_Pu_y_O_7_ are all larger than 0.25, as presented in Table 5, i.e., the Pu-substituted Gd_2_Zr_2_O_7_ exhibit good ductility. 

Elastic anisotropy is an important parameter for phase transformations, dislocation dynamics and geophysical applications [56]. Ranganathan and co-workers [57,58] proposed the universal elastic anisotropic index to indicate the elastic anisotropy of cubic crystals. The index $A^{U}=5\frac{G^{V}}{G^{R}}+\frac{B^{V}}{B^{R}}-6$ is investigated for all the considered compositions., where $A^{U}=0$ describes an isotropic crystal [57,58]. The calculated $A^{U}$ for both Gd_2−y_Pu_y_Zr_2_O_7_ and Gd_2_Zr_2−y_Pu_y_O_7_ are shown in Table 5. For Gd_2_Zr_2_O_7_, our calculated value of 0.01355 is comparable with other calculated value of 0.00420 [53]. As Pu is immobilized at Gd-site, we find that the $A^{U}$ values for all the compositions are nearly zero, indicating that the Gd_2−y_Pu_y_Zr_2_O_7_ compounds are isotropic elastically. As for the immobilization of Pu at Zr-site, the $A^{U}$ values are 0.21353 for Gd_2_Zr_1.0_Pu_1.0_O_7_ and 0.10062 for Gd_2_Zr_0.5_Pu_1.5_O_7_, indicative of elastic anisotropy.

The thermal properties of materials can be analyzed by the Debye temperature [4]. Table 5 lists the calculated and available experimental Debye temperature. The calculated $\theta_{D}$ value of 580.2 K for Gd_2_Zr_2_O_7_ is larger than the experimental result of 513.3 K [20], but shows better agreement with experimental value than another calculated value of 612.9 K [53]. As the Pu concentration increases, the $\theta_{D}$ value decreases for both Gd_2−y_Pu_y_Zr_2_O_7_ and Gd_2_Zr_2−y_Pu_y_O_7_. In particular, the $\theta_{D}$ values for Gd_2_Zr_2−y_Pu_y_O_7_ are smaller than those for Gd_2−y_Pu_y_Zr_2_O_7_. These results imply that the Gd_2_Zr_2−y_Pu_y_O_7_ compositions have a lower melting point and weaker interatomic binding force than the Gd_2-y_Pu_y_Zr_2_O_7_ compositions. Zhao et al. [59] calculated the Debye temperature for Th immobilization at Gd-site and Zr-site of Gd_2_Zr_2_O_7_ and observed similar phenomena, i.e., the Th-substituted Gd_2_Zr_2_O_7_ have a lower Debye temperature and especially smaller Debye temperature can be obtained by the Th immobilization at Zr-site than that at Gd-site.

The total and projected density of state (DOS) distributions for Gd_2−y_Pu_y_Zr_2_O_7_ and Gd_2_Zr_2−y_Pu_y_O_7_ are illustrated in Figure 6 and Figure 7, respectively. Table 1 and Table 2 list the band gap values. It is shown that all the compositions have large band gap values. For Gd_2_Zr_2_O_7_, the calculated band gap is 2.86 eV. For Pu incorporation into Gd-site, the band gap values of 2.27–2.37 eV are smaller than that of Gd_2_Zr_2_O_7_ and nearly independent of the Pu content. For Gd_2_Zr_2−y_Pu_y_O_7_, the changes in the band gap is more significant, ranging from 2.33 to 1.68 eV. It is indicated that Pu immobilization at Gd-site and Zr-site has different influences on the electronic structure of Gd_2_Zr_2_O_7_. The same conclusion can be drawn from the density of state distribution. For Gd_2−y_Pu_y_Zr_2_O_7_, O 2p orbital dominates and hybridizes with very few Gd 5d, Pu 5f and Zr 4d orbitals in the energy range of −5–1 eV, and the Pu 5f orbitals hybridize with the O 2p orbitals at the bottom of valence band. For Gd_2_Zr_2−y_Pu_y_O_7_, the O 2p orbital dominates and hybridizes with Pu 5f orbitals and very few Zr 4d and Gd 5d orbitals at the valence band. Obviously, in Gd_2_Zr_2_O_7_, different electronic structures can be obtained from the immobilization of Pu at Gd and Zr sites. These different electronic structures may result in discrepancies in the thermo-physical properties of Gd_2−y_Pu_y_Zr_2_O_7_ and Gd_2_Zr_2−y_Pu_y_O_7_.

## 4. Conclusions

The mechanical and electronic properties of Pu-containing Gd_2_Zr_2_O_7_ are studied by a DFT+U method. For Gd_2−y_Pu_y_Zr_2_O_7_ and Gd_2_Zr_2−y_Pu_y_O_7_, the elastic stability criteria are satisfied for all the calculated elastic constants, i.e., all the compounds are mechanically stable. As Pu immobilizes at Gd-site in Gd_2_Zr_2_O_7_, because the bonding distance and covalency of <Gd-O> and <Pu-O> bonds are comparable to each other, the elastic constants, elastic moduli, elastic isotropy and Debye temperature of Gd_2_Zr_2_O_7_ are all affected a little. As for Gd_2_Zr_2−y_Pu_y_O_7_, the elastic constants and elastic moduli change remarkably as compared with Gd_2_Zr_2_O_7_. The substitution of Pu for Zr sites increases the ionicity and decreases the elastic moduli, because the <Zr-O> bonds exhibit weaker ionicity than <Pu-O> bonds. In addition, the Debye temperature is decreased and the band gap is greatly reduced. Our calculations suggest that the Gd_2_Zr_2_O_7_ is a promising material for immobilizing nuclear waste such as Pu, while the thermo-physical of Gd_2_Zr_2_O_7_ may be influenced significantly after nuclear waste incorporation. 

## Figures and Tables

**Figure 1 nanomaterials-09-00196-f001:**
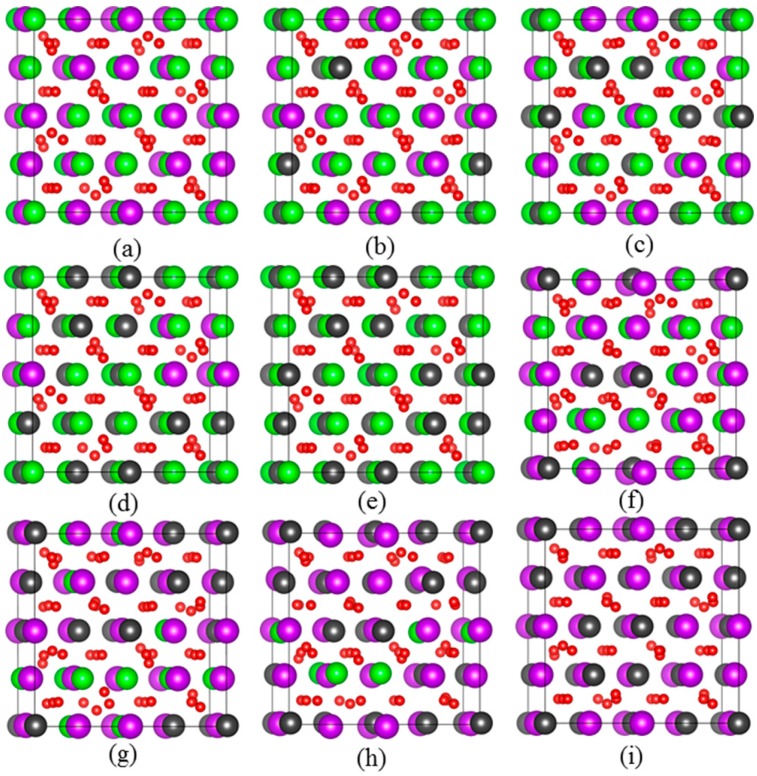
Schematic views of optimized geometrical structures for: (**a**) Gd_2_Zr_2_O_7_; (**b**) Gd_1.5_Pu_0.5_Zr_2_O_7_; (**c**) Gd_1.0_Pu_1.0_Zr_2_O_7_; (**d**) Gd_0.5_Pu_1.5_Zr_2_O_7_; (**e**) Pu_2_Zr_2_O_7_; (**f**) Gd_2_Zr_1.5_Pu_0.5_O_7_; (**g**) Gd_2_Zr_1.0_Pu_1.0_O_7_; (**h**) Gd_2_Zr_0.5_Pu_1.5_O_7_; (**i**) Gd_2_Pu_2_O_7_. The purple, dark grey, green and red spheres represent Gd, Pu, Zr and O atoms, respectively. (For interpretation of the references to color in this figure legend, the reader is referred to the web version of this article.)

**Figure 2 nanomaterials-09-00196-f002:**
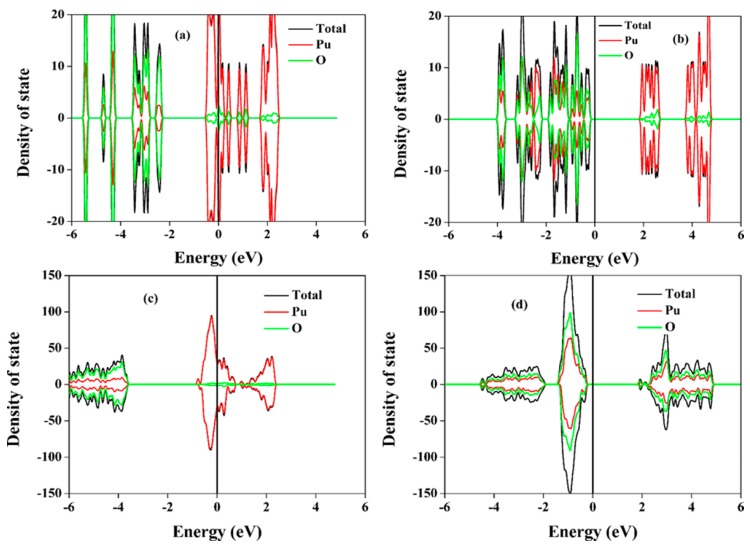
Density of state distribution for PuO_2_ at (**a**) U_eff_ = 0 eV and (**b**) 4 eV and for Pu_2_O_3_ at (**c**) U_eff_ = 0 eV and (**d**) 4 eV obtained by GGA+U.

**Figure 3 nanomaterials-09-00196-f003:**
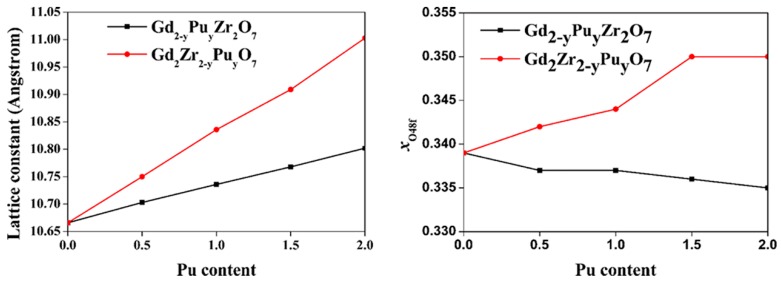
Variation of (**a**) lattice constant and (**b**) $x_{O48f}$ for Gd_2−y_Pu_y_Zr_2_O_7_ and Gd_2_Zr_2−y_Pu_y_O_7_ with Pu content. The calculated and fitted results are represented by symbols and dashed lines, respectively.

**Figure 4 nanomaterials-09-00196-f004:**
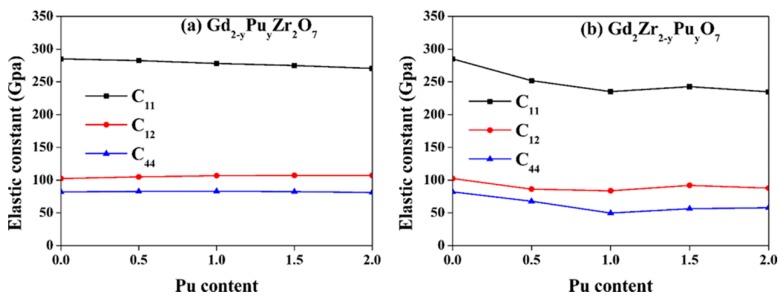
Variation of elastic constants (C_11_, C_12_ and C_44_) for (**a**) Gd_2−y_Pu_y_Zr_2_O_7_ and (**b**) Gd_2_Zr_2−y_Pu_y_O_7_ (0 ≤ y ≤2) with Pu content.

**Figure 5 nanomaterials-09-00196-f005:**
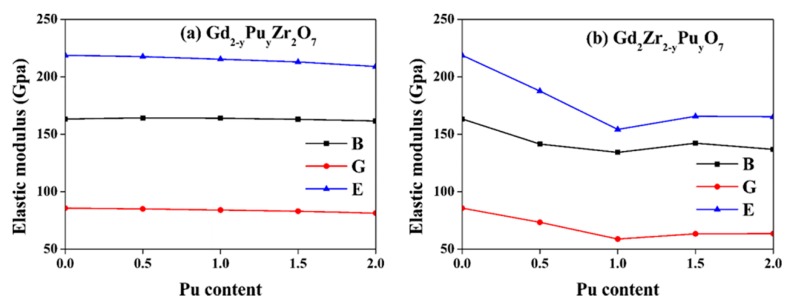
Variation of elastic moduli for (**a**) Gd_2−y_Pu_y_Zr_2_O_7_ and (**b**) Gd_2_Zr_2−y_Pu_y_O_7_ (0 ≤ y ≤2) as a function of Pu content. *B*: bulk modulus; *G*: shear modulus; *E*: Young’s modulus.

**Figure 6 nanomaterials-09-00196-f006:**
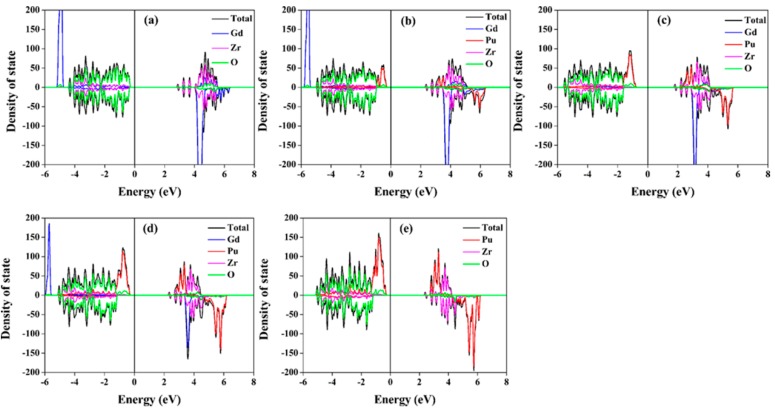
Total and projected density of state distributions for: (**a**) Gd_2_Zr_2_O_7_; (**b**) Gd_1.5_Pu_0.5_Zr_2_O_7_; (**c**) Gd_1.0_Pu_1.0_Zr_2_O_7_; (**d**) Gd_0.5_Pu_1.5_Zr_2_O_7_; (**e**) Pu_2_Zr_2_O_7_.

**Figure 7 nanomaterials-09-00196-f007:**
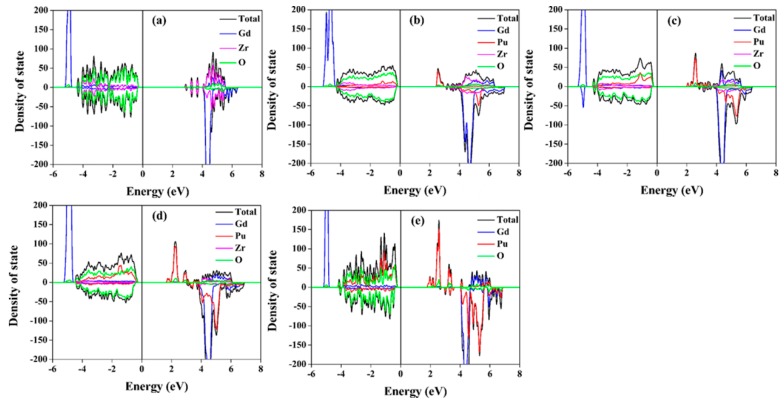
Total and projected density of state distributions for: (**a**) Gd_2_Zr_2_O_7_; (**b**) Gd_2_Zr_1.5_Pu_0.5_O_7_; (**c**) Gd_2_Zr_1.0_Pu_1.0_O_7_; (**d**) Gd_2_Zr_0.5_Pu_1.5_O_7_; (**e**)Gd_2_Pu_2_O_7_.

**Table 1 nanomaterials-09-00196-t001:** Lattice constant a_0_ (Å), O48f positional parameter *x* and bond distances (Å) for Gd_2−y_Pu_y_Zr_2_O_7_. Eg represents the band gap.

	a_0_	*x* _*O*48*f*_	E_g_ (eV)	d<Gd-O_48f_>	d<Gd-O_8b_>	d<Pu-O_48f_>	d<Pu-O_8b_>	d<Zr-O_48f_>
y = 0	10.666	0.339	2.86	2.553	2.309	-	-	2.109
Exp.	10.540 [38]	0.345 [41]		2.483 [38]				
Cal.	10.66 [18]	0.339 [18]		2.548 [18]	2.307 [18]			2.110 [18]
y = 0.5	10.703	0.337	2.33	2.561	2.302	2.582	2.369	2.111
y = 1.0	10.736	0.337	2.27	2.578	2.284	2.591	2.366	2.114
y = 1.5	10.768	0.336	2.33	2.574	2.286	2.605	2.349	2.116
y = 2.0	10.802	0.335	2.37	-	-	2.615	2.339	2.117
Exp.	10.70 [39]							

**Table 2 nanomaterials-09-00196-t002:** Lattice constant a_0_ (Å), O_48f_ positional parameter *x* and bond distances (Å) for Gd_2_Zr_2−y_Pu_y_O_7_. E_g_ represents the band gap.

	a_0_	*x* _*O*48*f*_	Eg (eV)	d<Gd-O_48f_>	d<Gd-O_8b_>	d<Pu-O_48f_>	d<Zr-O_48f_>
y = 0	10.666	0.339	2.86	2.553	2.309	-	2.109
Exp.	10.540 [38]	0.345 [41]		2.483 [38]			
Cal.	10.66 [18]	0.339 [18]		2.548 [18]	2.307 [18]		2.110 [18]
y = 0.5	10.750	0.342	2.33	2.513	2.338	2.296	2.110
y = 1.0	10.836	0.344	1.99	2.555	2.347	2.228	2.114
y = 1.5	10.909	0.350	1.68	2.508	2.364	2.273	2.121
y = 2.0	11.003	0.350	1.75	2.552	2.382	2.234	-

**Table 3 nanomaterials-09-00196-t003:** Elastic constants (C_11_, C_12_, C_44_, in GPa), bulk modulus *B* (GPa), shear modulus *G* (GPa), Young’s modulus *E* (GPa) of Gd_2−y_Pu_y_Zr_2_O_7_ and Gd_2_Zr_2−y_Pu_y_O_7_ (0≤ y ≤2).

		C_11_	C_12_	C_44_	*B*	*G*	*E*
Gd_2_Zr_2_O_7_		285.1	102.5	82.1	163.4	85.7	218.8
	Cal [43]	289	103	85	165	88	224
	Exp [45,46]				153	80	205
	Exp [20]				174	92	236
Gd_1.5_Pu_0.5_Zr_2_O_7_		282.6	105.1	82.7	164.3	85.1	217.6
Gd_1.0_Pu_1.0_Zr_2_O_7_		278.1	106.9	83.1	164.0	84.1	215.4
Gd_0.5_Pu_1.5_Zr_2_O_7_		274.9	107.2	82.5	163.1	83.0	213.0
Pu_2_Zr_2_O_7_		270.6	107.3	81.2	161.7	81.4	209.1
	Cal [2]	306	131.8	90.2			
Gd_2_Zr_1.5_Pu_0.5_O_7_		251.9	86.3	67.7	141.5	73.4	187.7
Gd_2_Zr_1.0_Pu_1.0_O_7_		235.2	83.8	49.8	134.3	58.9	154.2
Gd_2_Zr_0.5_Pu_1.5_O_7_		242.8	92.0	56.5	142.3	63.4	165.7
Gd_2_Pu_2_O_7_		234.8	87.9	57.8	136.9	63.6	165.3

**Table 4 nanomaterials-09-00196-t004:** Bader charge (|e|) for each ion in Gd_2−y_Pu_y_Zr_2_O_7_ and Gd_2_Zr_2−y_Pu_y_O_7_ (y = 0, 0.5, 1.0, 1.5, 2.0).

	Gd	Pu	Zr	O_48f_	O_8b_
Gd_2_Zr_2_O_7_	2.16	-	2.26	−1.25	−1.37
Gd_1.5_Pu_0.5_Zr_2_O_7_	2.15	2.10	2.27	−1.25	−1.35
Gd_1.0_Pu_1.0_Zr_2_O_7_	2.13	2.11	2.27	−1.24	−1.35
Gd_0.5_Pu_1.5_Zr_2_O_7_	2.15	2.09	2.28	−1.24	−1.33
Pu_2_Zr_2_O_7_	-	2.08	2.27	−1.23	−1.32
Gd_2_Zr_1.5_Pu_0.5_O_7_	2.14	2.36	2.26	−1.25	−1.37
Gd_2_Zr_1.0_Pu_1.0_O_7_	2.15	2.31	2.27	−1.25	−1.37
Gd_2_Zr_0.5_Pu_1.5_O_7_	2.15	2.34	2.24	−1.25	−1.36
Gd_2_Pu_2_O_7_	2.16	2.30	-	−1.26	−1.38

**Table 5 nanomaterials-09-00196-t005:** Pugh’s indicator (B/G), elastic anisotropy index ($A^{U}$), sound wave velocity ($v_{m}$, in m/s), Debye temperature (θ, in K) and Poisson’s ratio (ν) of Gd_2−y_Pu_y_Zr_2_O_7_ and Gd_2_Zr_2−y_Pu_y_O_7_ (0 ≤ y ≤ 2).

		B/G	$A^{U}$	$v_{m}$	θ	ν
Gd_2_Zr_2_O_7_		1.907	0.01355	4666.0	580.2	0.277
	Exp. [45,46]	1.913				0.276
	Exp. [20]	1.891			513.3	0.274
	Cal. [53]	2.004	0.00420	4833.5	612.9	0.286
	Cal. [43]					0.273
Gd_1.5_Pu_0.5_Zr_2_O_7_		1.931	0.00598	4533.7	560.7	0.279
Gd_1.0_Pu_1.0_Zr_2_O_7_		1.950	0.00105	4367.8	540.2	0.281
Gd_0.5_Pu_1.5_Zr_2_O_7_		1.964	0.00032	4247.4	522.5	0.282
Pu_2_Zr_2_O_7_		1.987	0.00004	4106.4	503.8	0.285
Gd_2_Zr_1.5_Pu_0.5_O_7_		1.928	0.04881	4124.6	508.7	0.279
Gd_2_Zr_1.0_Pu_1.0_O_7_		2.278	0.21353	3591.7	439.5	0.309
Gd_2_Zr_0.5_Pu_1.5_O_7_		2.243	0.10062	3590.2	435.9	0.306
Gd_2_Pu_2_O_7_		2.151	0.06923	3471.3	418.3	0.299

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
