# Peer review of "First-Principles Study of Thermo-Physical Properties of Pu-Containing Gd_2_Zr_2_O_7"

_nanomaterials, 2019, doi:10.3390/nano9020196_

Reviewer 1 Report

This paper is well written and deserves publication in my opinion. It contains the complete study of elastic properties which can be determined using VASP code. The problem studied by the authors is important and theoretical studies are very important, as the experimental investigation of Pu compounds is very difficult. Summarizng I recommend publication of that paper.

Author Response

Thank you very much.

Reviewer 2 Report

This is a well-written manuscript which I am happy to recommend for publication in Nanomaterials.

Author Response

Thank you very much.

Reviewer 3 Report

This is an interesting paper that summarizes a hard computational work.

Results look sound and the work deserves publication. However, I recommend the paper

for publication after the authors take into account the following:

Clearly show what is the role of the U correction (i.e., what is the difference between U=0 and U=4eV results, and how sensitive are the results with respect to the value of U).

Give argument that DFT+U approximation is sufficient, since it is known for example that the best approach for Pu is DFT+DMFT (see e g. Nature 410, 793 (2001)).

Author Response

Comment 1: Clearly show what is the role of the U correction (i.e., what is the difference between U=0 and U=4eV results, and how sensitive are the results with respect to the value of U).

Response: As Pu substitutes for Gd3+ and Zr4+ in Gd2Zr2O7, the corresponding valence states for Pu are Pu3+ and Pu4+, respectively.  Because in both PuO2 and Pu2O3 the Pu 5f electrons are strongly correlated, Hubbard U correction is thus necessary. In the revised manuscript, we present the density of state distribution for both PuO2 and Pu2O3 at Ueff =0 eVand Ueff =4 eV in Fig. 2. It is shown that without Hubbard U correction, i.e., at Ueff =0 eV, the Pu 5f electrons are itinerant and delocalized over the Fermi lever, resulting in metallic states. At Ueff =4 eV, the Pu 5f electrons are localized and the system becomes insulating, which is consistent with the experimental finding. Hence, Hubbard U correction is considered in this work.

In J. Nucl. Mater. 467 (2015) 937, Zhao et al. investigated the effects of Ueff value on the density of state, lattice constant, volume, band gap and reaction energy for both PuO2 and Pu2O3 and compared with experimental values. They reported that the lattice constant, volume, and band gap increase with the Ueff value and the reaction energy decreases with the Ueff value. In our work, a Ueff =4 eV for Pu in both PuO2 and Pu2O3 yield lattice constants and band gap that are in good agreement with experimental values. Hence, a Ueff =4 eV is employed in this work.

Corresponding modifications have been made in page 3, lines 86-93 in the revised manuscript.

 Comment 2: Give argument that DFT+U approximation is sufficient, since it is known for example that the best approach for Pu is DFT+DMFT (see e g. Nature 410, 793 (2001)).

Response: The calculated lattice constant of 5.46 Å for PuO2 and 11.18 Å for Pu2O3 obtained at Ueff =4 eV are comparable to the experimental values of 5.39 Å (Journal of Applied Physics. 2013, 113) and 10.98 Å (J. Nucl. Mater. 1964, 12, 131-141), respectively. The calculated band gap for Pu2O3 when Ueff =4 eV is 1.757 eV, which correspond to the experimental value of 2 eV (Physical Review B. 2008, 78). Thus, DFT+U approximation is sufficient, and we use Ueff =4 eV in our calculations for Pu immobilization in Gd2Zr2O7. On the other hand, the Ueff =4 eV is also consistent with the value of 4-5 eV that are reported in the literature (Chemical Reviews. 2013, 113, 1063-1096).

Corresponding discussions have been made in page 3, lines 93-95 in the revised manuscript, and page 4, lines 96-98.

Reviewer 4 Report

The incorporation of Pu into Gs_2Zr_2O_7 pyrochlore is studied theoretically, using Density Functional Theory plus U method. The authors analyzed structural parameters, elastic moduli,
elastic isotropy, Debye temperature, Bader charges, and density of states.
The manuscript is interesting, and may be important for the investigation of nuclear waste immobilization. However, I can not recommend the publication because of the following
problems:
- it is not clear to me what GGA exchange-correlation (XC) functional was used in the calculations. Ref. [25] shows a GGA hole model that can be used for PW91 and PBE functionals. However, there are recent XC functionals that are more accurate than PW91/PBE for solid state calculations, such
as Phys. Rev. B 91, 041120(R) (2015), Phys. Rev. B 73, 235116 (2006), or Phys. Rev. B 93, 045126 (2016). This modern XC development, at least, can be acknowledged.
- a short discussion about the value of U should be provided
- how important are relativistic effects for such systems, and for the studied properties?
This problem should be carefully explained.
- Tables 4 and 5 are missing. The manuscript should be updated.

- overall, this work is close to the one of Ref. [18]. The authors should better stress the novelty of the present manuscript.
Author Response

 Comment 1: it is not clear to me what GGA exchange-correlation (XC) functional was used in the calculations. Ref. [25] shows a GGA hole model that can be used for PW91 and PBE functionals. However, there are recent XC functionals that are more accurate than PW91/PBE for solid state calculations, such
as Phys. Rev. B 91, 041120(R) (2015), Phys. Rev. B 73, 235116 (2006), or Phys. Rev. B 93, 045126 (2016). This modern XC development, at least, can be acknowledged.

Response: In this study, the PBE functional under GGA approximation is employed. Our calculations show that several properties for the investigated systems agree well with available experimental and theoretical data in the literature.

In the revised manuscript, the recent XC functionals all have been acknowledged, please see page 2, line 71.

 Comment 2: a short discussion about the value of U should be provided

Response: The discussions on the value of U have been made on page 3, lines 93-95 and page 4, lines 96-98.

 Comment 3: how important are relativistic effects for such systems, and for the studied properties?
This problem should be carefully explained.

Response: According to Einstein’s special relativity theory of 1905, the mass of any moving object increases with its speed, , where mrel, me, v and c {\displaystyle \displaystyle m_{e},v_{e},c}are the relativistic mass, electron rest massvelocity of the electron and speed of light, respectively. Arnold Sommerfeld calculated that, for a 1s electron of a hydrogen atom with an orbiting radius of 0.0529 nm, α ≈ 1/137, where α is the fine-structure constant. That is to say, the fine-structure constant shows the electron traveling at nearly 1/137 the speed of light (Journal of Chemical Education. 1991, 68, 110-113). In the density functional theory method, it is assumed that the velocity of electrons is much smaller than the speed of light, i.e., v << c. Correspondingly, mrel ≈ me. That is to say, the density functional theory, which mainly handle the motion of electrons at ground states, is mainly based on the non-relativistic effect and the relativistic effect is nealy negligible.

Comment 4: Tables 4 and 5 are missing. The manuscript should be updated.

Response: Tables 4 and 5 have been added to the manuscript.

 Comment 5: overall, this work is close to the one of Ref. [18]. The authors should better stress the novelty of the present manuscript. 

 Response: In Ref. [18], the authors mainly investigated the solution behaviour of Ce at Gd-site in Gd2Zr2O7, i.e., the incorporation energy and solution energy, etc. In our work, we mainly focus on how the incorporation of Pu at Gd- and Zr-site influences the mechanical properties of Gd2Zr2O7. The novelty of the current work has been stressed in the abstract.

Round  2

Reviewer 4 Report

The revised manuscript is significantly improved, and the authors responded to all the suggestions. Overall this work is important, and I recommended the publication.

However, I would like to tell to the authors, that DFT can properly take into account relativistic effects, there is available the full Dirac-Kohn-Sham theory, and any popular code, as VASP, contains relativistic effects for DFT calculations. See for example

J. Hafner, Ab‐initio simulations of materials using VASP: Density‐functional theory and beyond